# COVID-19 Lung Ultrasound Scores and Lessons from the Pandemic: A Narrative Review

**DOI:** 10.3390/diagnostics13111972

**Published:** 2023-06-05

**Authors:** Luigi Maggi, Giulia De Fazio, Riccardo Guglielmi, Flaminia Coluzzi, Silvia Fiorelli, Monica Rocco

**Affiliations:** 1Government of Italy Ministry of Interior, 00189 Rome, Italy; 2Department of Medical-Surgical Sciences and Translational Medicine, Sapienza University of Rome, 00189 Rome, Italy; 3Department of Medical and Surgical Sciences and Biotechnologies, Sapienza University of Rome, 04100 Latina, Italy; 4Unit of Anaesthesia, Intensive Care and Pain Medicine, Sant’Andrea University Hospital, 00189 Rome, Italy

**Keywords:** lung ultrasound, score, COVID-19

## Abstract

The WHO recently declared that COVID-19 no longer constitutes a public health emergency of international concern; however, lessons learned through the pandemic should not be left behind. Lung ultrasound was largely utilized as a diagnostic tool thanks to its feasibility, easy application, and the possibility to reduce the source of infection for health personnel. Lung ultrasound scores consist of grading systems used to guide diagnosis and medical decisions, owning a good prognostic value. In the emergency context of the pandemic, several lung ultrasound scores emerged either as new scores or as modifications of pre-existing ones. Our aim is to clarify the key aspects of lung ultrasound and lung ultrasound scores to standardize their clinical use in a non-pandemic context. The authors searched on PubMed for articles related to “COVID-19”, “ultrasound”, and “Score” until 5 May 2023; other keywords were “thoracic”, “lung”, “echography”, and “diaphragm”. A narrative summary of the results was made. Lung ultrasound scores are demonstrated to be an important tool for triage, prediction of severity, and aid in medical decisions. Ultimately, the existence of numerous scores leads to a lack of clarity, confusion, and an absence of standardization.

## 1. Introduction

Recently, WHO declared that COVID-19 no longer constitutes a public health emergency of international concern [1]. However, pathology remains an ongoing health issue to be dealt with during clinical practice, just like other infectious diseases.

COVID-19 still has a wide range of clinical presentations, from asymptomatic cases to severe cases requiring intensive treatment due to respiratory failure and multi-organ dysfunction [2].

In response to the challenges posed by the COVID-19 pandemic, several protocols and scoring systems have been developed to diagnose and redirect clinical judgment. Lung ultrasound (LUS) has emerged as a suitable alternative to CT scans due to its practicality, low cost, lack of radiation risk, and minimal requirement for health personnel [3]. The peripheral distribution of COVID-19 in the lungs makes ultrasound particularly well-suited to investigate the disease [4]. Moreover, ultrasound findings have been shown to be compatible with CT findings [5,6]. However, ultrasound provides non-specific signs that can be related to other respiratory diseases commonly encountered in emergency/critical settings [7,8], so interpretation of results must be done within the clinical context. It is also worth noting that the interpretation of ultrasound is operator-dependent and requires specific training [9].

The popularity of LUS has led to the development of grading systems to assign numerical values to each feature found during an ultrasound examination. Even prior to the COVID-19 pandemic, lung ultrasound scores were developed to create a common language among operators for diagnosis and clinical decision making [10,11,12]. However, the pandemic led to the readaptation of existing scores to the unique characteristics of COVID-19 pneumonia and the need to reduce the risk of spreading the infection, resulting in the creation of various scoring systems that lack standardization and cause confusion [13].

As COVID-19 enters a new phase and health systems return to pre-pandemic normalcy, the disease is no longer in the spotlight. Therefore, it is important to simplify the diagnosis and treatment process for COVID-19 in an easier and standardized way, as it is becoming just one of many important medical conditions encountered in daily practice. Nevertheless, the experience gained during the pandemic should not be set aside.

There have been numerous scores created to diagnose and treat COVID-19 during the pandemic. The aim of this study is to clarify the key aspects of lung ultrasound scores to standardize their clinical use in a non-pandemic context.

## 2. Methods

Authors performed a search on PubMed for papers relating to “COVID-19”, “ultrasound”, and “Score” until 5 May 2023.

The searching strategy comprehends terms such as “thoracic”, “lung”, “echography”, and “diaphragm”. Together, researchers evaluated the bibliography for the most relevant articles. Due to limited data, performing a statistical analysis was not feasible and the results of the search were summarized discursively.

## 3. Results

During the pandemic, several LUS scores were developed, either as revised versions of existing scores for respiratory diseases or as new scores. All these scores are based on the same principles of lung ultrasound, which are combined in different ways to create a numerical score that is integrated with other ultrasound findings.

### 3.1. Lung Findings

A-Lines: horizontal hyperechoic static lines, periodically spaced from the pleura, indicating a normal ventilated lung [14]. They are ultrasound reverberations of the pleural line and indicate a fully aerated lung.

B-Lines (Figure 1): comet-like artefacts, vertical linear lines which move concurrently with lung sliding, starting from the pleural line up to the edge of the screen, erasing the A-lines [15]. The presence of an increasing number of B-lines on ultrasound is directly proportional to a more serious stage of interstitial pathology from a moderate loss of aeration to a complete loss of aeration [16]. When the number of B-lines becomes greater than 3 or when they converge, we can appreciate the “white lung” on LUS, associated with the finding of ground glass opacities at CT scan [17].

Pleural effusions (Figure 2): present at LUS as hypo or anechoic regions between parietal and visceral pleura [18]. When scanned with M-Mode, the “sinusoid sign” can be appreciated, which is due to the motion of the floating lung in the pleural effusion fluid [19]. Transudates are usually homogeneous and anechoic, while exudates may appear heterogeneous and loculated [20].

Pleural irregularities: The pleural line is seen as a hyperechoic line moving synchronously with breath in a fully aerated lung; in a normal adult it is located 2.5 cm from the skin [21]. Pleural irregularities are characterised as the disappearance of the typical hyperechoic pleural profile [22]. The pleural line may be discontinuous and present an increase in thickness.

Consolidations (Figure 2): occur when air normally contained in the alveoli is substituted by material of diverse origin. When they are located near the pleura, the acoustic impedance normally seen between pleura and aerated parenchyma is reduced, so that consolidations may be morphologically studied [23]. Consolidations may be translobar which appears as a tissue-like echostructure similar to the one of the liver (hence, it is called hepatization), while non-translobar appears with irregular margins between the consolidation and the fully aerated lung [24]. Consolidations at ultrasound may show air bronchograms that appear as either hyperechoic branching and tubular structures or hyperechoic millimetre long multiple structures [25]. They are the analogues of the bronchograms seen in chest X-rays.

### 3.2. Other Findings

Diaphragmatic findings (Figure 3): ultrasound study of the diaphragm provides a non-invasive, feasible, and dynamic method to evaluate the movement of the diaphragm together with its characteristics such as thickness [26,27,28]. To evaluate the diaphragm, the probe is positioned below the costal margin either at the midclavicular line or at the anterior-axillary line. The evaluation may be done either at the left or right hemithorax taking advantage of the acoustic window provided by the spleen or liver, respectively. The diaphragm is detected through the two-dimensional mode (2D); then, M-mode is used to evaluate the movement of the structure. Moreover, ultrasound assesses the excursion, the speed of diaphragmatic contraction, the inspiratory time, and the duration of a respiratory cycle, but also the diaphragmatic thickness [29], which is a proxy of diaphragmatic power [30].

The standing drawback is that artefacts visualization strongly depends on the machine settings (in particular frequency) and the probe utilized and the interpretation of what is seen is strongly operator dependent: actual protocols do not provide a strict quantitative measure to define what is and what is not an artefact and for the evaluation of pleural irregularities and consolidations [31].

For this reason, is of paramount importance the development of standardized practice as it exists for other anatomical regions.

### 3.3. COVID-19 Characteristics at LUS

The ultrasound appearance of COVID-19 is consistent with its pathophysiological basis as an inflammatory interstitial disease that gradually impacts the alveoli and reduces aeration.

Typical findings in COVID-19 pneumonia include a B-line pattern (either focal or diffuse), a thickened or irregular pleural line, and consolidations of various sizes ranging from small subpleural to large translobar [6]. The most common finding is B-lines (shown in Figure 1), which may fuse together to create the characteristic “white lung” appearance. B-lines serve as a densitometer, indicating the progressive loss of air-filled alveoli [3]. Typically, the findings are patchy and have a bilateral distribution, with clusters alternating with spared areas, leading to the definition of a “storm of clusters of B-lines” [31]. Other ultrasound findings in COVID-19 pneumonia include thickening and irregularities of the pleural line, consolidations, and less commonly, pleural effusions [14]. Consolidations are more commonly seen in severe and critical patients, particularly in the posterior fields [32]. Lung parenchyma consolidation indicates complete alveolar de-aeration and corresponds to a more serious stage of the illness.

### 3.4. Lung Ultrasound Scores for Predicting Severity, Treatment Response, and Outcomes

The scores presented in Table 1 demonstrated to be useful in predicting the severity of disease, the need for oxygen support, NIMV, and evaluating whether the support chosen has been efficient. The statistical data and advantages of each score are presented in Table 2.

Quantitative LUS score (q-LUS) and coalescent LUS score (c-LUS) were extensively used in the pre-COVID-19 era to assess lung aeration [10,11]. In the setting of COVID-19 pneumonia, c-LUS has been found to strongly correlate with Chest-CT [32] and is useful for assessing the benefits of recruiting maneuvers and changes in ventilation settings [33]. It can also predict outcomes in asymptomatic frail patients [34]. Falgarone et al. proposed an index with 89% sensitivity and 100% specificity in predicting an abnormal CT exam: a value of 0.32 was set as a threshold for the need for oxygen support [35]. Conversely, Soldati et al. proposed a lung ultrasound protocol specific to COVID-19 [36] with a high negative predictive value [37]. The score showed to be associated with patient worsening in medium to low-intensity care units (AUC 0.82), with the need for high-flow oxygen support, ICU admission, and death [38]. A higher total score is associated with pleural effusions, a lower P/F ratio, and higher lactate dehydrogenase (LDH) [38].

When combined with coagulation parameters and compressive ultrasound (CUS), the Soldati protocol proved to be useful in evaluating the length of hospitalization and the need for O2 therapy [39]. Notably, the Soldati score demonstrated being a good predictor of fatality (AUC 0.878, sensitivity 87.5%, specificity 81.7%), and helpful in discerning whether patients would benefit from HFNC or MV [40].

The use of lung ultrasound scores to predict the appropriateness and efficacy of NIV support has generated considerable interest. The Integrated LUS score (I-LUS) [41] has been shown to be effective in distinguishing patients who would benefit from oxygen therapy alone versus those who require NIV. Patients redirected to intensive care units had higher I-LUS scores compared to those redirected to low-intensity units. Similarly, Casella et al. proposed a score to predict the need for Continuous Positive Airway Pressure ventilation, which was predictive of death and transfer to the ICU [42].

A simplified LUS score was proposed for the early assessment of the lung as a predictor of NIV support failure (defined as the death of the patient or the need for endotracheal intubation) in the first 24 h [43]. The authors found that a score >11 is associated with worsening clinical outcome and admission to ICU in 72 h, while a score >12 and ≥5 areas involved are indicative of NIV failure (sensitivity of 88%, specificity of 93%).

Interestingly, using the same score it was found a significant association between simplified LUS and hypercoagulability state largely known to be a cornerstone of COVID-19 pathophysiology [44] to have a robust correlation with disease severity [45].

Dargent et al. proposed a score in which a rating of 27 is related to extubation failure and prompt need for NIV support [46]. Conversely, a reduction of the modified LUS score is associated with successful extubation [47].

**Table 1 diagnostics-13-01972-t001:** Lung Ultrasound scoring systems and corresponding scanning areas on thorax.

**Coalescent Lung score c-LUS** [10,11]	Score 0: presence of A-lines, maximum 2 B-lines Score 1: ≥3 well-spaced B-lines Score 2: coalescent B-lines Score 3: tissue-like pattern	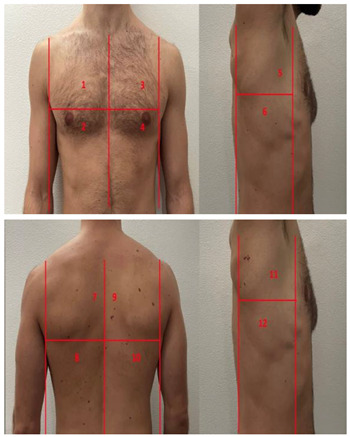 12 AREAS
**Quantitative LUSS (q-LUS)** [12]	Score 0: A-lines, maximum 2 B-lines Score 1: artefacts occupying ≤ 50% of pleura Score 2: artefacts occupying > 50% of pleura Score 3: tissue-like pattern	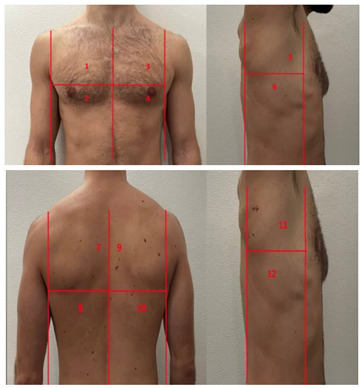 12 AREAS
**Soldati Score** [36]	Score 0: continuous and regular pleural line, presence of A-lines. Score 1: indented pleural line, with the presence of vertical white areas. Score 2: broken pleural line with the appearance of small-to-large consolidations associated with white lung. Score 3: dense and largely extended white lung with or without consolidations.	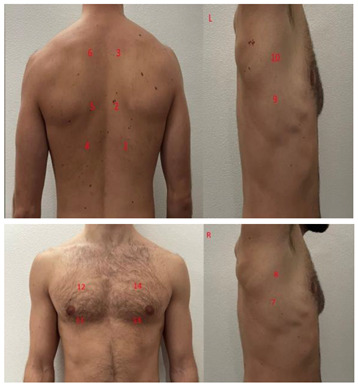 14 AREAS
**Falgarone index** [35]	1: normal pleural image with A-lines 2: B lines 3: Multiple B-lines (ground glass) 4: consolidations 5: neo organisation known as hepatisation 6: pleurisy (not found among the patients) Index: a score from 1 to 5 is given to each area, the scores obtained are added up together. The resulting number is divided to the maximal score possible considering only the evaluated areas.	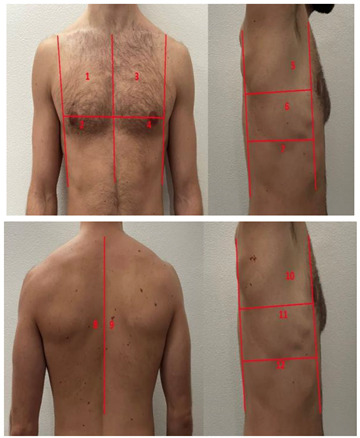 12 AREAS
**Simplified LUS score** [43]	1: a small loss of aeration characterised by more than three B-lines or presence of multiple sub-pleuric consolidations separated by normal pleura; 2: a moderate loss of aeration consisting of multiple and coalescent B-lines and/or multiple sub-pleuric consolidations 1 × 2 cm or smaller and separated by thickened or irregular pleura; 3: a severe loss of aeration described as parenchymal consolidation or subpleuric consolidations greater than 1 × 2 cm.	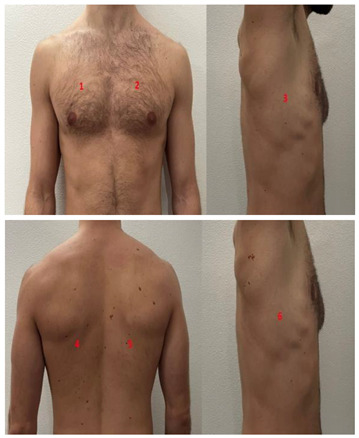 6 AREAS
**Integrated ultrasound score (I-LUS)** [41]	0: A-lines or ≤2 B-lines plus regular sliding 1: ≥3 B lines or spaced focal points plus regular sliding 2: coalescing B-lines 3: pulmonary consolidations Plus: Presence of pleural effusion (1: present, 0: absent). Presence of pericardial effusion (1: present, 0: absent). Measurement of the IVC respiratory variation (<0–33%) (1: present, 0: absent). Diaphragm excursion: measured in normal respiration, with M-mode through a right subcostal scan. A value > 2 +/− 0.5 cm is considered normal (0 points), while an inferior value is considered abnormal (1 point).	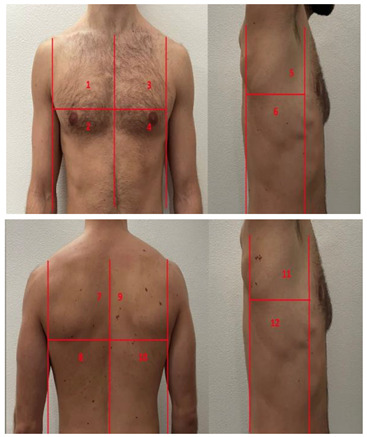 12 AREAS
**Casella score** [42]	0: regular pleural line, presence of horizontal artefacts (A-lines); 1: at least 3 B-lines in at least one scan of the region; the B-lines do not merge one in the other. Small subpleural consolidations ≤1 cm diameter may be present; 2: multiple, converging B-lines, usually determining a so-called “white lung”. Small subpleural consolidations ≤1 cm diameter may be present; 3: presence of at least one consolidation with major axis >1 cm.	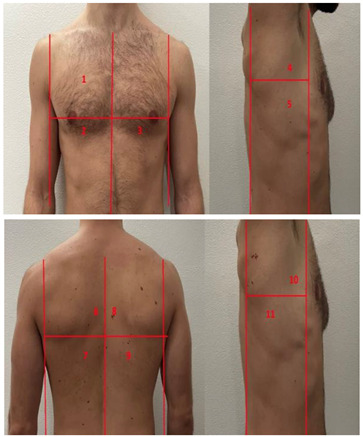 11 AREAS
**Dargent score** [46]	0 = normal findings 1 = well-defined B-lines 2 = coalescent B-lines 3 = consolidations	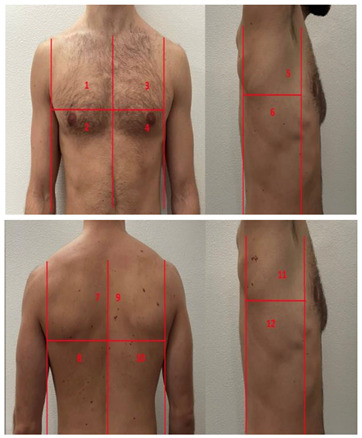 12 AREAS
**Modified LUS score** [47]	0: well-spaced B-lines <3 1: well-spaced B-lines ≥3 2: multiple coalescent B-lines 3: lung consolidation Plus, plural line is quantitatively ranked as follows: 0: normal 1: irregular pleural line 2: blurred pleural line	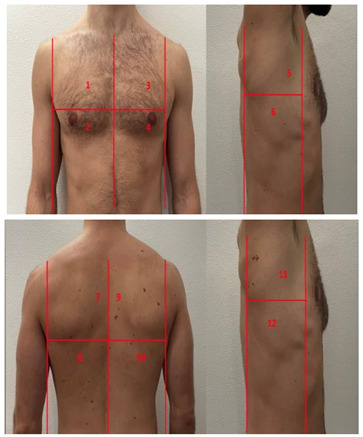 12 AREAS

Several of these scores have been correlated with other laboratory or clinical parameters to increase their utility in clinical settings [39,48]. Notably, Boero et al. developed the “COVID-19 Worsening Score” (COWS) [49]. A LUSS >15, in combination with four other variables (P/F ratio, dyspnea at admission, number of comorbidities, and days from symptom onset), is useful in estimating a patient’s risk of disease progression and can discriminate those at high or low risk of worsening, with an overall accuracy of 80% and a negative predictive value of 93% [49].

**Table 2 diagnostics-13-01972-t002:** Advantages and performance measures of lung ultrasound scores.

	Soldati Score	Falgarone Index	Simplified LUS Score	Integrated LUS Score	Casella Score	C-LUS
**Worsening predictor**	AUC = 0.82			Good correlation *p* < 0.001		
**Diagnosis**	Sensitivity 99% Specificity 56%					Sensitivity 100% Specificity 40%
**Correlation with CT scan**		Sensitivity 90%Specificity 100%				Positive correlation *p* < 0.0001
**Oxygen requirement**		Sensitivity 95% Specificity 67%				
**NIV requirement/NIV failure**			Sensitivity 88% Specificity 93%	Positive correlation *p* = 0.005	Positive correlation *p* < 0.0001	
**Outcome**	Sensitivity 87% Specificity 82%			Positive correlation *p* = 0.005	Positive correlation *p* = 0.005	Positive correlation *p* < 0.05
**Advantages**	Specific for Covid-19	High correlation with CT-scan	Fast to perform	Takes into account IVC, Diaphragm and Pericardium	Valuable prognostic tool in hospitalized patients	High sensitivity for asymptomatic patients

Sensitivity and specificity scores are represented. In case of missing data, the correlation between the topic and scores has been provided by supplying either the value of p or the AUC (area under the curve) value.

## 4. Discussion

The massive engorgement of emergency departments during the pandemic has prompted the usage of lung ultrasound as a point-of-care methodology to dispatch patients in wards with different levels of intensities [50,51], highlighting the importance of lung ultrasound for triage. It can predict the severity of COVID-19, guide treatment decisions, and detect the need for respiratory support [14,43,52,53,54].

The implementation of scores to stratify COVID-19 has obvious advantages from an economic standpoint, avoiding unnecessary hospitalization and redirecting intensive support. The goal of applying scoring systems is to translate an ensemble of ultrasonographic qualitative features and patterns into numbers [55]. Efforts have been made towards this goal during and even before the COVID-19 pandemic, resulting in a myriad of scores that often bear little differences between each other. However, the establishment of new scores and revision of pre-existing scoring systems have led to a lack of clarity. We speculate that disarrayed information may pose a difficulty for those who are first approaching the technique. Moreover, in our opinion, this may increase the lack of standardization and inter-operator variability which is already a main limitation of the methodology.

An issue that has raised a great deal of interest is the number of areas that should be scanned. In COVID-19, where the lung findings are preferentially located in the posterior regions [56], an accurate evaluation of these areas may be advantageous. However, a common field should be defined to address underestimation related to the limitation of inspected areas [57,58,59].

Various studies have been conducted with the aim of finding the appropriate number of areas to be scanned in COVID-19-affected patients. From their results, a scanning protocol including 12 zones is comparable to the commonly used protocol of 14 zones [56,60,61]. When considering a 10-area protocol [60] it seems feasible only if it considers posterior and basal areas, according to the already stressed knowledge of the preferential posterior distribution of COVID-19-related lung damage. However, it is not recommended to scan fewer than 10 areas as this would lead to underestimation of the damage. [61]. Some protocols with less than 10 areas were proposed [43], which bear the advantage of reducing the spread of infection during examinations.

On the other hand, even more extensive protocols have been proposed. It has been suggested that an 18-zone protocol would be more accurate if performed with the patient in a lateral decubitus position [62]. The authors suggest that this protocol would allow for a more extensive evaluation of the thorax, a simultaneous evaluation of anterior lateral and posterior regions, and the lateral decubitus position helps to reduce the gravity-related confounding effect on lung aeration [62].

It is important to emphasize the evaluation of posterior regions in COVID-19 patients. Soldati et al. recommend the observation of three posterior regions in their protocol, but it may not always be possible for the patient to maintain a sitting position [36]. In such cases, the authors suggest evaluating postero-basal regions instead.

However, Casella et al. demonstrated that a score obtained using only anterior and lateral areas is still a reliable predictive tool, significantly associated with respiratory failure progression [42]. This may be useful when the patient’s position is limited, and posterior areas are difficult to assess.

Another important issue is the orientation of the probe. Scans are traditionally performed longitudinally, allowing for easy identification of the pleural line, but the visualization of the parenchyma may be limited by the size of the intercostal space.

The reliability of a lung ultrasound (LUS) score can be limited when it is based on the number of artefacts seen, which is why a transverse approach is preferred to evaluate lung aeration [12]. To address this issue, a quantitative LUS score was developed [12], which focuses on the percentage of pleura involved rather than the presence of B-Lines, as was used in the previously developed coalescent LUS score. By considering the amount of pleura involved, nonhomogeneous diseases such as ARDS, ventilator-associated pneumonia (VAP), and lung contusion can be better evaluated for the severity of loss of aeration. The intercostal approach is recommended in the Soldati score [36] and the modified LUS score [47], where not only whether the pleura is involved but also its appearance (i.e., indented or broken in the Soldati score, irregular or blurred in the modified LUS score) is considered. In particular, the modified LUS score evaluates the parenchyma and the pleura separately, which may provide an advantage in prognostic value. Serial m-LUS score evaluations were found to be more sensitive than LUS score evaluations for predicting the need for prolonged mechanical ventilation. It is important to note that, unlike imaging of other organs, lung parenchymal ultrasound does not rely on the direct visualization of anatomical landmarks, and its use is based on the interpretation of artefacts. Therefore, the visualization of artefacts is highly operator dependent, and features obtained from non-optimized settings could over or under-estimate the severity of illness. This issue is amplified in an emergency setting, such as during a pandemic. Consequently, clear definitions for features should be decided, as qualitative descriptions are highly susceptible to personal interpretation and may lead to confusion. The same considerations should be adopted when areas are graded.

Regarding lung ultrasound scoring, the Soldati score [36] proposes a grading system that avoids counting B lines, as their enumeration may not be a reliable parameter due to differences in probe and imaging settings [36,63]. Instead, the presence of lung consolidations (also known as a tissue-like pattern) is often used to indicate more severe stages of the disease. However, the lack of sizing for consolidations can lead to an overestimation of loss of aeration, particularly in non-homogeneous pathologies such as ARDS [64,65]. Some authors have proposed cut-offs and rankings for consolidations in their scores [42,43,46], but a clear and standardized definition is still lacking.

As COVID-19 continues to be encountered in the clinical setting alongside other pathologies, clear and standardized protocols for diagnosis and clinical decision making should be established. Efforts towards these goals have been made [66], but a consensus-based clinical tool that is easily implemented should be considered mandatory.

## 5. Limitations

A limitation of the study is that the scores mentioned pursue different outcomes, so results are not easily comparable to each other.

## 6. Conclusions

During the pandemic, lung ultrasounds gained considerable popularity. Many authors worked to reduce the main limit of lung ultrasound as operator dependence and lack of specificity. Scores were proposed, but inevitably this created confusion and a lack of standardization. Additional research and a consensus seem mandatory for a standardized approach to lung ultrasound for COVID-19 as the pandemic has finally ended.

## Figures and Tables

**Figure 1 diagnostics-13-01972-f001:**
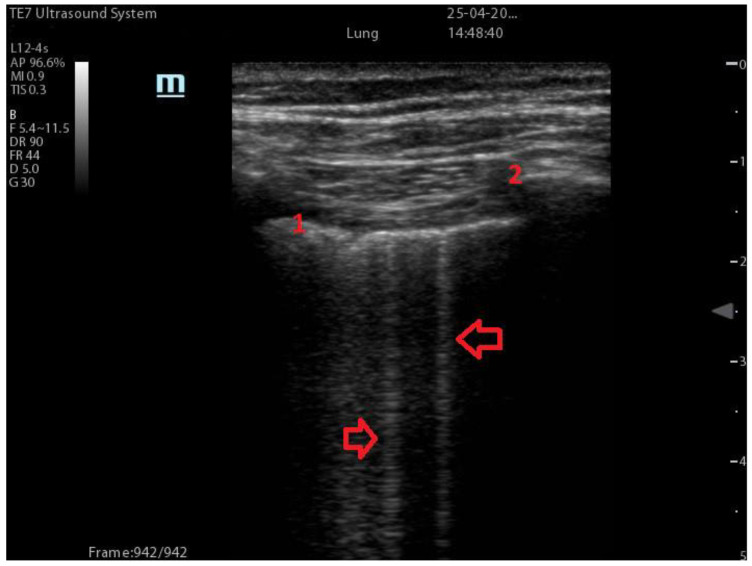
Number 1 indicates the pleural line. Number 2 indicates the rib and the rib shadow. Arrows indicate single B-lines.

**Figure 2 diagnostics-13-01972-f002:**
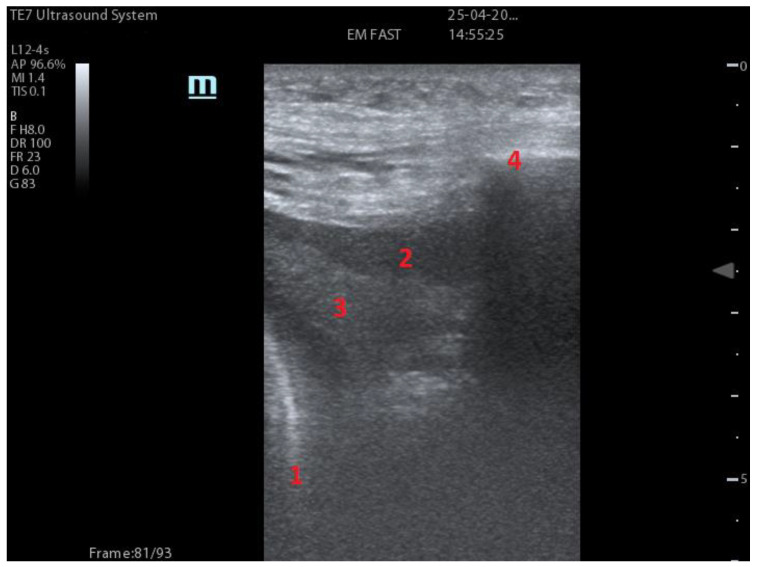
Number 1 indicates the diaphragm. Number 2 indicates pleural effusion. Number 3 indicates consolidated lung. Number 4 indicates the rib and the rib shadow.

**Figure 3 diagnostics-13-01972-f003:**
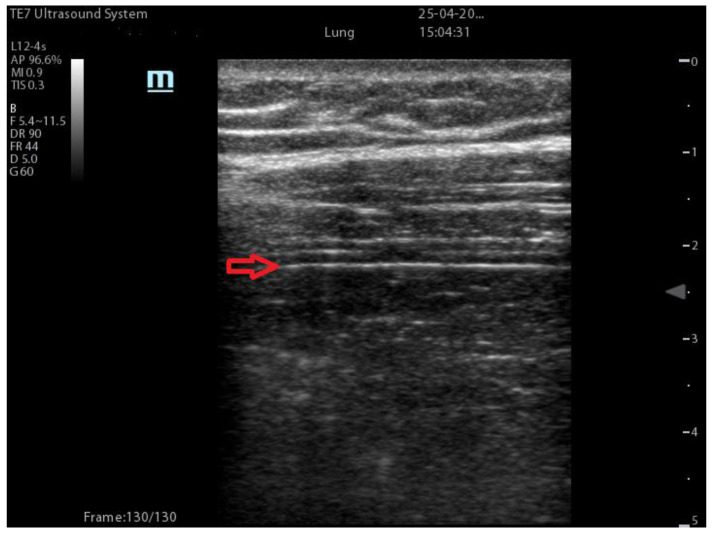
The red arrow indicates the diaphragm.

## Data Availability

Not applicable.

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
