# Peer review of "COVID-19 Lung Ultrasound Scores and Lessons from the Pandemic: A Narrative Review"

_diagnostics, 2023, doi:10.3390/diagnostics13111972_

Round 1
Reviewer 1 Report (Previous Reviewer 2)
Dear authors
The paper has been improved. But I suggest to add a table, that shows the results of each score, regarding accuracy, specifcity and sensitivity and provides pros and cons.
Ad that stage it is just a summary of the "how to do" of the different methods, but I miss a comparison among the scores also regarding the value.
Author Response
Dear Collegue, we thanks you for the efforts in optimizing our paper. As advised, we provided a table with advantages, sensibility and specificity data when available. We remain open to any further advice that can lead to improve the article.
Thanks

Reviewer 2 Report (Previous Reviewer 1)
Thank you for rewriting the manuscript according to the reviewers comments.
The manuscript improved and should be concidered for publication.
Author Response
Dear collegue. We wish to thank you for all the advices in the process.
This manuscript is a resubmission of an earlier submission. The following is a list of the peer review reports and author responses from that submission.
Round 1
Reviewer 1 Report
The manuscript reviews lung ultrasound findings in covid-19 patients with a focus on scoring methods. The diagnostic accuracy of Lung ultrasound scores, were discussed and differences addressed with a strong recommendation for harmonisation.
This work is of great importance for the clinical lung ultrasound community and for the clinical covid management. The manuscript is well written with proper English and mostly structured according to the authors guidelines. However, it has some limitations regarding reporting of introduction and discussion as well as a missing method section, therefore it should be considered for publication after revision.
General comments:
The title of the manuscript is pointing to the usability of lung ultrasound score for clinicians. The manuscript presents LUS scores found in literature but not so much their use in clinical practice. Therefore the title to me not entirely reflect the content ”COVID-19 lung ultrasound scores, a narrative review” or the value of the scores should be better outlined.
Introduction section.
An overview of diagnostic COVID-19 pneumonia monitoring is given. The value of lung ultrasound is highlighted. However, the introduction section is roughly clustered in single sentences by paragraphs. This way of presenting information is hard to read and guide to the main motivation. The binding information between these paragraphs is missing. Please reformulate to make the introduction easier readable.
Methods section is missing describing the way of literature research As well as summarising the overall findings of the review research.
Findings:
The first sentence Paige 2 -line 744. During the pandemic…. Is a valuing Of LUS protocols and belongs to the discussion section.
Page 2 line 91 The sentence about the B lines origin is not conclusive please reformulate.
Page 3 line 93 pleural Effusions should be bold
Inferior vena cava and pericardial effusion. On page 3 are not generally examined during lung ultrasound. They are not mentioned in the scores, therefore it is questionable to present here as a lung ultrasound feature. Consider deleting.
Page 4 line 137 lung parenchyma ultrasound…. Such sentence belongs to the discussion section.
Page 4 line 156-8 an image would be helpful understanding this paragraph.
When comparing the LUS scores I would recommend to implement a table where are all the scores can be easy overseen and compared on one page.
Figures: The figures in this review are only related to the scan areas. I would recommend implementing all images into one figure scheme, where the regions are horizontal ordered, for easier overseeing the changes. An maybe add figures of specific LUS artefacts.
Pitch 7 line 259: Is it condensation or consultation? Ground glass rocket Is a completely new phrase to me please stick with the international terminology such as white lung.
Page 10 line 386: Several authors have proposed modified scores. For instance, reference 16 Gutsche et al Proposed the same score for hospitalised covid patients.
Finally, I miss a statistical valuing of the presented scores based on diagnostic accuracy or receiver operator Curve (AUC) based on the existing literature.
Discussion:
Similar to the introduction section, the chapter is a rough alignment of statements in paragraphs. This is very hard to read. I would recommend reformulating and taking more time in guiding the reader to the main content of the work.
When working with LUS Scores, the main difference is related to summarising all scores for each areal or taking the maximum score found on the patient’s chest. Such different approaches found in literature should be more intense discussed because of impact on clinical indication / diagnostic needs.
The limitation section is missing.
Author Response
The manuscript reviews lung ultrasound findings in covid-19 patients with a focus on scoring methods. The diagnostic accuracy of Lung ultrasound scores, were discussed and differences addressed with a strong recommendation for harmonisation.
This work is of great importance for the clinical lung ultrasound community and for the clinical covid management. The manuscript is well written with proper English and mostly structured according to the authors guidelines. However, it has some limitations regarding reporting of introduction and discussion as well as a missing method section, therefore it should be considered for publication after revision.
General comments:
The title of the manuscript is pointing to the usability of lung ultrasound score for clinicians. The manuscript presents LUS scores found in literature but not so much their use in clinical practice. Therefore the title to me not entirely reflect the content ”COVID-19 lung ultrasound scores, a narrative review” or the value of the scores should be better outlined.
Introduction section.
An overview of diagnostic COVID-19 pneumonia monitoring is given. The value of lung ultrasound is highlighted. However, the introduction section is roughly clustered in single sentences by paragraphs. This way of presenting information is hard to read and guide to the main motivation. The binding information between these paragraphs is missing. Please reformulate to make the introduction easier readable.
Methods section is missing describing the way of literature research As well as summarising the overall findings of the review research.
Findings:
The first sentence Paige 2 -line 744. During the pandemic…. Is a valuing Of LUS protocols and belongs to the discussion section. Done as requested
Page 2 line 91 The sentence about the B lines origin is not conclusive please reformulate. The sentence has been deleted
Page 3 line 93 pleural Effusions should be bold Done as requested
Inferior vena cava and pericardial effusion. On page 3 are not generally examined during lung ultrasound. They are not mentioned in the scores, therefore it is questionable to present here as a lung ultrasound feature. Consider deleting. Deleted as requested
Page 4 line 137 lung parenchyma ultrasound…. Such sentence belongs to the discussion section. Done as requested line 441 page 11.
Page 4 line 156-8 an image would be helpful understanding this paragraph. Image inserted
When comparing the LUS scores I would recommend to implement a table where are all the scores can be easy overseen and compared on one page. Done as requested (table1)
Figures: The figures in this review are only related to the scan areas. I would recommend implementing all images into one figure scheme, where the regions are horizontal ordered, for easier overseeing the changes. And maybe add figures of specific LUS artefacts.
Pitch 7 line 259: Is it condensation or consultation? Ground glass rocket Is a completely new phrase to me please stick with the international terminology such as white lung. Done as requested
Page 10 line 386: Several authors have proposed modified scores. For instance, reference 16 Gutsche et al Proposed the same score for hospitalised covid patients. Done in line 234 page 6
Finally, I miss a statistical valuing of the presented scores based on diagnostic accuracy or receiver operator Curve (AUC) based on the existing literature. Done as requested.
Discussion:
Similar to the introduction section, the chapter is a rough alignment of statements in paragraphs. This is very hard to read. I would recommend reformulating and taking more time in guiding the reader to the main content of the work.
When working with LUS Scores, the main difference is related to summarising all scores for each areal or taking the maximum score found on the patient’s chest. Such different approaches found in literature should be more intense discussed because of impact on clinical indication / diagnostic needs.
we revised introduction and discussion
The limitation section is missing. Done as requested.
Reviewer 2 Report
In the present paper, the authors describe different scores regarding the usefulness of ultrasound examination in CoVID 19 patients. The authors conclude that there is no adequate score and that the scores lead to more confusion than standardisation.
This paper has aspects of an educational paper as well as that of a review paper. In this sense, the findings described in great detail at the beginning show little connection with CoVID 19 pneumonia. The various scores described afterwards are listed in great detail. However, there is no clear overview in the form of a table or diagram that allows the benefits of the various scores to be weighed against each other. On the one hand, the scores differ in their indications, in the questions they are asked and also in their outcomes. It is therefore not surprising that the individual scores are not comparable with each other. However, the work does not allow any comparison at all and it requires a clear comparison. Also, although various pictures are given regarding the place where the examination was carried out, there is no picture information regarding the findings. The authors put a lot of effort into compiling the data, but lack any sense for the reader.
Although I agree with the authors' conclusion that ultrasound does not show any significant benefit in the context of scores, there were certain positive aspects of ultrasound in the context of visualising infiltrates.
Author Response
Dear reviewer,
we revised our paper following your instructions. Images and tables have been inserted to hopefully aid the reader in the understanding of the text. Also a limitation section has been inserted.
Kind regards
Round 2
Reviewer 1 Report
The Authors have responded to most of the aspects mentioned in the first report. However, the majour aspect making of readable for the user such as presenting the scores in a table and reducing the images regarding scaned areas were not performed. Therefore I can only advice to revise the manuscript again.
Table 1 is missing
Fig of LUS artefacts are missing
Limitation schould adress the circumstance that each score is mainly invented under different clinical purposes (screening, ICU, progress) and ist therefore not comparable in a direct measure of diagnostic acouracy.
Reviewer 2 Report
Dear Author
in my first review I wrote the following sentences:
However, there is no clear overview in the form of a table or diagram that allows the benefits of the various scores to be weighed against each other. On the one hand, the scores differ in their indications, in the questions they are asked and also in their outcomes. It is therefore not surprising that the individual scores are not comparable with each other. However, the work does not allow any comparison at all and it requires a clear comparison. Also, although various pictures are given regarding the place where the examination was carried out, there is no picture information regarding the findings. The authors put a lot of effort into compiling the data, but lack any sense for the reader.
So far Im not aware, that these remarks have been implemented (for example I do not see any table).